# Therapies beyond Physiological Barriers and Drug Resistance: A Pilot Study and Review of the Literature Investigating If Intrathecal Trastuzumab and New Treatment Options Can Improve Oncologic Outcomes in Leptomeningeal Metastases from HER2-Positive Breast Cancer

**DOI:** 10.3390/cancers15092508

**Published:** 2023-04-27

**Authors:** Oana Gabriela Trifănescu, Dan Mitrea, Laurenția Nicoleta Galeș, Ana Ciornei, Mihai-Andrei Păun, Ioana Butnariu, Raluca Alexandra Trifănescu, Natalia Motaș, Radu Valeriu Toma, Liviu Bîlteanu, Mirela Gherghe, Rodica Maricela Anghel

**Affiliations:** 1Department of Oncology, “Carol Davila” University of Medicine and Pharmacy, 020021 Bucharest, Romania; 2Department of Radiotherapy II, “Prof. Dr. Al. Trestioreanu” Institute of Oncology, 022328 Bucharest, Romania; 3Neuroaxis Neurology Clinic, 011302 Bucharest, Romania; 4Department of Medical Oncology II, “Prof. Dr. Al. Trestioreanu” Institute of Oncology, 022328 Bucharest, Romania; 5Department of Neurology, National Institute of Neurology and Neurovascular Diseases, 041914 Bucharest, Romania; 6Discipline of Endocrinology, “Carol Davila” University of Medicine and Pharmacy, 011863 Bucharest, Romania; 7“C. I. Parhon” Institute of Endocrinology, 020021 Bucharest, Romania; 8Department of Thoracic Surgery, “Prof. Dr. Al. Trestioreanu” Institute of Oncology, 020021 Bucharest, Romania; 9Department of Radiotherapy I, “Prof. Dr. Al. Trestioreanu” Institute of Oncology, 022328 Bucharest, Romania; liviu.bilteanu@gmail.com; 10Department of Nuclear Medicine, “Prof. Dr. Al. Trestioreanu” Institute of Oncology, 022328 Bucharest, Romania

**Keywords:** intrathecal treatment, leptomeningeal metastasis, breast cancer, intrathecal trastuzumab

## Abstract

**Simple Summary:**

Approximately 10% of HER2-positive breast cancer patients will develop leptomeningeal metastases (LM), characterized by the spread of tumor cells within the leptomeninges and subarachnoid space. Historically, patients with HER2-positive breast cancer and LM have been excluded from studies regarding anti-HER2 therapies, and as such, the data on this topic are scarce. This pilot study evaluated the efficacy of local treatment with intrathecal Trastuzumab (IT) added to systemic treatment. The oncologic outcome of 14 patients with HER2-positive LM is reported. Seven patients received IT, and seven received the current standard of care (SOC). The intrathecal administration of Trastuzumab alongside systemic treatment and radiotherapy improves oncologic outcomes in LM HER2-positive breast cancer with manageable toxicity.

**Abstract:**

Leptomeningeal metastases (LM) are a rare but rapidly fatal complication defined by the spread of tumor cells within the leptomeninges and the subarachnoid space, found in approximately 10% of patients with HER2-positive breast cancers. This pilot study evaluated the efficacy of local treatment with intrathecal Trastuzumab (IT) added to systemic treatment. The oncologic outcome of 14 patients with HER2-positive LM is reported. Seven received IT, and seven received standard of care (SOC). The mean number of IT cycles administered was 12.14 ± 4.00. The response rate to CNS after IT treatment + SOC was 71.4%, and three patients (42.8%) obtained durable responses lasting more than 12 months. The median progression-free survival (mPFS) after LM diagnosis was six months, and the median overall survival (mOS) was ten months. The mean values of the PFS in favor of IT therapy (10.6 mo vs. 6.6 mo) and OS (13.7 vs. 9.3 mo) suggest a non-negligible investigation direction in the sense of exploiting intrathecal administration as a possible treatment modality in these patients. Adverse events reported were local pain related to intrathecal administration and one case of arachnoiditis, hematoma, and CSF fistulae. Intrathecal administration of Trastuzumab, alongside systemic treatment and radiotherapy, might improve oncologic outcomes in LM HER2-positive breast cancer with manageable toxicity.

## 1. Introduction

The leptomeninges consists of the arachnoid and the pia mater membrane. The space separating these two membranes is the subarachnoid space which contains cerebrospinal fluid [1]. Leptomeningeal metastasis (LM) represents the infiltration of cancer cells into the cerebrospinal fluid and leptomeninges [2].

The most frequent causes of LM are breast cancer (11% to 64%), lung cancer (14% to 29%), and melanoma (6% to 18%) [2]. Given the difference in incidence, it is not surprising that breast cancer patients (2.26 million new cases worldwide in 2020, according to the GLOBOCAN cancer statistics) account for most cases of LM [3,4]. The brain is becoming a predilect metastatic site in HER2-positive and triple-negative breast cancer, only brain metastasis (BM) originating from the lungs being more frequent [5,6].

Dissemination of cancer cells into the leptomeninges is a major complication that results in substantial morbidity and mortality, and without proper treatment, the median survival does not exceed 4–6 weeks [1]. After systemic treatment, the median overall survival (mOS) reaches 3–6 months, and even so, only 15–24% of patients survive for more than a year [5,7].

LM is confirmed by finding malignant cells on CSF analysis or using specific neuroimaging findings on gadolinium-enhanced MRI of the complete neuroaxis in a suggestive clinical context of a patient with advanced or metastatic cancer [7].

The typical finding on the MRI of an LM patient is the enhancement of the leptomeninges, cranial nerves, brain surface, cerebellar foliae, or spinal nerve roots. The enhancement may be linear, nodular, curvilinear, focal, or diffuse. Quantitative assessment is rarely possible due to the lesions’ small volume and complex geometry [8].

Positive CSF cytology is the gold standard diagnostic test for LM and can sustain a diagnosis based on clinical features even if the MRI scan is negative [9].

Unfortunately, many CSF specimens contain few malignant cells, making this a difficult pathologic diagnosis. The possibility of finding atypical cells increases by up to 90% with repeated lumbar punctions, with most cases requiring three to four CSF samples analyzed. False-negative findings are frequent, up to 40% [10], but false positives are rare, mainly due to the misinterpretation of reactive lymphocytes as malignant cells [2].

Biopsy of the leptomeningeal lesions is rarely indicated, mostly in cases with repeated negative CSF cytology or when lesions are identified on imaging without an identified primary tumor [2,7].

Staining neoplastic cells for specific alterations, such as the HER2-protein in breast cancer by immunochemistry, is useful in determining whether the molecular pattern from the primary tumor has changed.

Based on clinical findings, imaging, and identifying tumor cells in CSF, EANO-ESMO proposed a comprehensive flow-chart, dividing LM in type I (positive CSF cytology), type II (possible/probable when neurological symptoms are present, and CNS MRI confirms LM spread). MRI findings can be linear (sub-type A), nodular (sub-type B), both (sub-type C), or presenting features suggestive of hydrocephalus (sub-type D) [7]. A recent study applied this classification to 254 LM and found an important prognostic value [11].

There are few treatment options in patients with LM metastatic disease, and patients are generally excluded from clinical trials due to poor outcomes. Current expert opinions recommend focal radiotherapy in patients with symptomatic lesions and whole-brain radiation therapy (WBRT) for patients with extensive nodular and associated brain metastasis, all in association with current practice of anti-HER2 treatment. Current challenges and unmet needs in treating patients with HER 2-positive LM breast cancer are to preserve the quality of life by delaying neurological deterioration and to improve survival [12,13].

Trastuzumab pioneered anti-HER2 targeted therapy, against the extracellular domain of the HER2 protein. It has been hypothesized that its antitumor activity is enacted by activating the immune response via antibody-dependent cell-mediated cytotoxicity leading to an adaptive immune response or through downregulation of the intracellular pathway via PI3K and MAPK pathways [14,15,16,17,18].

The biggest issue for the systemic treatment of HER2-positive LM patients is the need to overcome the blood–brain barrier, which is impervious to molecules with a molecular weight higher than 400–500 Da, thus limiting the efficacy of systemic treatment [19]. Unfortunately, Trastuzumab cannot pass the blood-brain barrier, having a molecular weight of approximately 148 kDa [20].

Although the blood–brain barrier can be circumvented through intrathecal administration, the concurrent systemic anti-HER2 treatment must consider that the tumor cells can become resistant to Trastuzumab during the treatment. Incriminated mechanisms of resistance include increased signaling from other HER receptors (such as HER3 or epidermal growth factor receptor) [21]; structural modifications of the antibody binding site, leading to Trastuzumab binding impairment [22,23]; mutations in the HER2/ERBB2 gene (such as L755S) [24,25,26,27]; increased intratumoral heterogeneity of HER2 expression [28,29]; and increased activity and expression of drug efflux pumps [30,31,32].

As such, we set out to demonstrate that intrathecal administration of Trastuzumab improves oncological outcomes by circumventing an important physiological barrier to treat LMs while emphasizing that the choice of systemic therapy needs to consider overcoming the mechanisms of resistance that naturally arise during anti-HER2 targeted therapy.

## 2. Materials and Methods

This study evaluated the efficacy of local treatment with intrathecal Trastuzumab added to systemic treatment in patients with HER2-positive breast cancer and LM spread.

We present a prospective pilot study that included 14 patients with LM metastasis from HER2-positive breast cancer treated with standard of care plus/minus intrathecal Trastuzumab. Treatment management was done in strong collaboration between the oncologic department, radiotherapy department, and a neurologic facility and compared the results. The treatment decision was taken by a multidisciplinary team as often as possible.

The time interval for recruiting patients is between 2016 and 2022.

Inclusion criteria were HER2-positive breast cancer patients with imaging highly suggestive for LM on MRI, with or without CSF positive cytology, with Karnofsky performance status of more than 60, willing to undergo multimodality treatment and normal hematological, renal, and hepatic function.

The study was approved by the Local Ethical Committee of the Institute of Oncology (24935/2022), and all patients signed the institutional Informed Consent Form (ICF). The study was conducted in harmonization with the World Medical Association (WMA) Helsinki Declaration of 1975, as revised in 2008.

Trastuzumab was administered intrathecally via repeated lumbar puncture at 150 mg every three weeks, associated or not with intrathecal dexamethasone 2 or 4 mg. Descriptive statistics (mean and standard error, median, and standard deviation) were used to characterize the two groups. The Kolmogorov–Smirnov normality test has been applied to test the normal distribution across the two groups of quantitative variables (such as age, KPS, time to first diagnostic, time to brain tumor diagnostic, leptomeningeal tumor dissemination etc.). For the variables exhibiting normal distributions, Student *t*-tests have been applied to compare the means. In contrast, for all the others, we have applied non-parametric tests to compare the distributions (Mann–Whitney U-test, Kolmogorov–Smirnov test), to compare the means across groups (independent median test) and to estimate the confidence interval of median difference across the groups (Hodges–Lehmann). Since all the variables were not normally distributed, correlation coefficients were calculated using the Spearman scheme.

Through the group’s small dimensions, we attempted to classically evaluate the oncologic outcome for the LM patients using the Kaplan–Meier method to determine median progression-free survival (PFS) and overall survival (OS). PFS was defined as the time from LM diagnosis to the leptomeningeal disease progression on imaging or death from any cause. Overall survival was defined as the time from LM diagnosis to death due to any cause. The univariate analysis using the log-rank test was used to analyze the influence of different factors regarding the oncologic outcome. A multivariate analysis was used according to the stepwise Cox proportional hazards model to identify independent prognostic factors and estimate their effect on the time to disease progression and overall survival. *p* value was considered statistically significant if it was <0.05.

## 3. Results

### 3.1. Patient Characteristics

Patients diagnosed and treated between 2016 and 2022 in the Medical Oncology, Radiotherapy, and Neurology departments were included. The treatment decision was taken by a multidisciplinary team as often as possible. The median age of the patients was 54 years, with a mean ± standard error of 54.07 ± 2.58. Four patients were diagnosed with Stage III disease and underwent neoadjuvant treatment and surgery, and ten were diagnosed directly with stage IV disease. The median time since the initial diagnosis of breast cancer to brain metastasis was 35 months, and the median time since the initial diagnosis of breast cancer and LM was 43 months. One patient had only LM without evidence of brain metastasis. 

For the entire group of patients, the median Karnofsky Performance status (KPS) evaluated by the neurologist at the moment of LM diagnosis was 75, and the mean KPS was 78.57 ± 3.60.

The histopathologic report showed invasive ductal carcinoma in all patients, estrogen or progesterone receptor positivity in 10 patients (71.42%), and HER2 positivity (9 patients showed HER2 3+ on immunohistochemistry while 5 patients showed HER2 2+ and FISH positiveness). The mean value of Ki67 was 34.69 ± 10.57%.

A diagnosis of leptomeningeal disease was made according to the corroboration of clinical symptoms, MRI evaluation, and CSF analysis. In most cases, at diagnosis, the patients presented with neurological symptoms, the most notable of which were neuropathic pain, motor deficit, focal motor seizures, and facial hemiparesis. MRI showed pathognomonic images of LM. Type A (linear) alterations were observed in 4 patients, type B (nodular) in 2 patients, and Type C in 8 patients—none of the patients presented at the diagnostic with hydrocephalus. Cerebrospinal fluid analysis reveals neoplastic cells in 5 patients.

Patient characteristics for each subgroup (intrathecal treatment + standard of care or just standard of care) are summarized in Table 1 and Table 2.

There are no significant differences across groups regarding the median age, median KPS or median BM or LM diagnostic durations. This shows that using intrathecal Trastuzumab was not biased by the values of these variables. Moreover, the mean or median comparison tests showed that the groups did not exhibit a statistically significant difference in progression-free or overall survivals calculated concerning different events in patient evolution (first diagnosis, and BM and LM occurrence).

For all 14 patients, the KPS has positive and statistically significant correlation coefficients with the PFS (0.649, *p* = 0.011) and OS (0.547, *p* = 0.043) after LM. When looking only at intrathecal-treated patients, these correlation coefficients increase to 0.780 (*p* = 0.038) and 0.722 (*p* = 0.067), though the latter has no statistical significance. In the same group, age is positively correlated with the duration of the LM diagnostics (0.779, *p* = 0.039). KPS and age are, thus, evolution predictors of the treatment course.

### 3.2. Surgical Treatment

Four patients underwent brain surgery to eliminate brain metastasis, and two of those patients had leptomeningeal metastasis near the surgical cavity.

### 3.3. Radiotherapy

WBRT was administered to 11 patients (78.6%), and stereotactic radiotherapy (Gamma-Knife) was administered to 7 patients (50%). Craniospinal radiotherapy was administered to 4 (28.6%) patients.

### 3.4. Intrathecal Treatment

The mean number of IT cycles administered was 12.14 ± 4.00, range of 4 to 35, and the mean duration of IT therapy was 10.57 ± 3.16 months.

The response rate to CNS after IT treatment + SOC was 71.4%, and three patients (42.8%) obtained durable responses lasting more than 12 months.

### 3.5. Systemic Treatment

Systemic therapy (chemotherapy and targeted anti-HER2 treatment) was administered according to the ESMO Guidelines for treating HER2-positive disease at the time of disease progression. For the entire group of patients, the systemic treatment consisted of Pertuzumab + Trastuzumab + Chemotherapy in 3 (21.4%) patients, TDM-1 in 4 patients (28.6%), Lapatinib + Capecitabine in 5 patients (35.7%), and Tucatinib + Trastuzumab + Capecitabine in 2 patients (14.3%), both in the intrathecal group.

### 3.6. Oncological Outcomes

The breast cancer oncologic outcomes were estimated using Kaplan–Meier methods. The median follow-up since breast cancer diagnosis for the entire group of patients is 40.5 months. The median PFS (mPFS) in the first line for all 14 patients was 20 months, and the median overall breast cancer survival (mOS) was 52 months.

mPFS after LM diagnosis estimated with the Kaplan–Meier method for the entire lot of 14 patients was six months, and the mOS was ten months (Figure 1). The 1-year and 2-year-PFS were 32% and 10%, respectively, while the 1-year and 2-years-OS were 38% and 16%, respectively.

Though no statistically significant differences (treated vs. non-treated) have been found across the groups due to low dimension groups, the mean values of the PFS (10.6 mo vs. 6.6 mo) and OS (13.7 vs. 9.3 mo) suggest a non-negligible investigation direction in the sense of exploiting intrathecal administration as a possible treatment modality in these patients who are otherwise non-eligible for any standard treatment. This is a promising approach in the all-patient-encompassing concept of treatment personalization, which does not mean only palliative care.

Subgroup analysis showed that the mPFS after LM diagnosis for patients who received intrathecal Trastuzumab was 15 months compared to 6 months in patients who did not. In a multivariant Cox regression analysis, the addition of intrathecal Trastuzumab was associated with a decreased risk of progression (HR = 0.38, 95% CI 0.09–1.531, *p* = ns). The mOS for LM patients was 8 months in patients with the standard of care and 22 months in patients who received standard of care and intrathecal Trastuzumab. Intrathecal Trastuzumab plus standard of care was associated with a statistically significant and clinically meaningful benefit in reducing the risk of death by LM (HR = 0.198, 95% CI 0.041–0.961, *p* = 0.045) compared to the standard of care. The percent of patients free of progression at one year was 16% in SOC and 56% in IT. The percent of patients alive in the SOC group was 16% vs. 64% in IT.

In our lot of patients, there was no difference regarding PFS and OS in patients receiving WBRT and craniospinal radiotherapy. There was a statistical difference in patients receiving SBRT vs. no SBRT (mPFS 4 vs. 15 months), but data may be confounded by the fact that patients receiving SBRT had a lower tumor burden.

Regarding systemic therapy, the median estimated PFS was 6 months for Pertuzumab + Trastuzumab + Chemotherapy, 4 months for TDM-1, 8 months for Lapatinib + Capecitabine, and 18 months for Tucatinib + Trastuzumab + Capecitabine. mOS, according to treatment, was eight months for TDM-1, ten months for Lapatinib + Capecitabine, ten months for the Pertuzumab + Trastuzumab combination, and not reached for Tucatinib.

The response was evaluated using a complete neurologic examination and MRI. In Figure 2 and Figure 3, MRI scans of the patients showing long-lasting response and rapid responses are displayed.

The treatment was associated with manageable side effects. All patients reported pain at the puncture site (100%); one patient (14.3%) presented with orthostatic hypotension, one patient (14.3%) presented with subdural hematoma, one patient (14.3%) presented with CSF fistulae, and one patient presented arachnoiditis and needed intrathecal corticosteroids.

## 4. Discussion

Despite important advances in oncologic breast cancer treatment (in radiotherapy, chemotherapy, targeted therapy, and surgery), LM still has poor prognostic due to significant neurologic morbidity and mortality [33].

Instituting a standard treatment regimen for LM is challenging due to the relative rarity of these patients, the rapid progression of the disease, and the need for clinical trial inclusion. LM remains an exclusion criterion in most clinical trials. Even today, most evidence supporting treatment procedures comes from retrospective studies, small prospective studies, post-hoc analyses, and meta-analyses [34].

Systemic disease control in HER2+ breast cancer patients has greatly improved since the introduction of anti-HER2 therapies, with Trastuzumab being the backbone of treatment for this subgroup of patients. However, with better survival rates comes a higher incidence of CNS metastases, including LM (6.8% risk of developing brain metastasis in 10 years). The diagnosis of LM has a devastating impact on the course of the disease, with a great reduction in the quality of life and neurologic dysfunction that impairs the ability of the patient to function independently and significantly lowers the survival rates [35,36].

There are limited treatment options for HER2+ LM, and there is a need to find new efficient therapeutic agents and a stronger collaboration between specialists to improve these patients’ prognosis and quality of life.

The current standard of care for LM management is multidisciplinary treatment, including radiotherapy (RT), systemic and intrathecal (IT) chemotherapies plus targeted therapies, and surgery (palliative ventriculoperitoneal shunting for increased intracranial pressure) [3].

Intrathecal treatment is achieved by administering therapeutic agents directly into the subarachnoid space. This increases the CSF’s drug concentration, inducing tumor cell necrosis. Subsequently, it is the most common way to deliver chemotherapeutic drugs in linear and non-bulky LM [37].

Intrathecal administration of chemotherapeutic agents has proved a good approach for patients presenting with LM. However, using harsh chemotherapeutic medications such as Methotrexate, Cytarabine, and Thiotepa, administered alone or in combination with hydrocortisone, is not without side effects, with many patients presenting with aseptic meningitis [2]. Although there are good results in preventing LM caused by leukemia and lymphoma, the role of these agents in solid tumors with LM is limited because they have a narrow antitumor activity spectrum [37]. Only one study found the addition of intrathecal liposomal cytarabine to systemic treatment as an improvement in leptomeningeal metastases–progression-free survival (PFS) (3.8 vs. 2.2 months) [38].

One of the main challenges is the lack of efficacy of the anti-Her2 agents in the intracranial disease because IV Trastuzumab has little to no penetration through the BBB (25–50% of cases will develop brain metastases and 6–7% will develop LM). Moreover, although the BBB can be affected by the local disease, and CNS irradiation is known to increase the barrier’s permeability, Trastuzumab still cannot reach therapeutic concentrations in the CSF [39]. In pharmacokinetics studies, Trastuzumab concentration in the CSF was 300 times lower than its concentration in the serum after IV administration [14]. Because it delivers an active medication to the location of the disease, intrathecal delivery of Trastuzumab is a desirable option for LM patients.

Multiple phase I/II trials and case reports have studied the intrathecal administration of Trastuzumab, providing a good insight into the safety and efficacy that it may provide. There has yet to be a universal agreement on the proper dosage and frequency of intrathecal Trastuzumab. However, the most common schedules used in the scientific literature range from 5 to 150 mg twice weekly, weekly, or every three weeks. Intrathecal Trastuzumab appears safe and has no known major side effects, whether alone or in combination therapy [36,40,41].

Figura et al. reported 13 cases of LM treated with IT in 5 years, an mPFS of 5.6 months, and sustained more than 6 months of response in 2 patients [42].

One multicenter phase I/II trial has analyzed the safety and efficacy of intrathecal administration of Trastuzumab. It has been determined that intrathecal Trastuzumab can be safely administered with a dose of up to 80 mg twice weekly. The results showed stable disease in 50% of patients and partial response in 19% of cases, with a median overall survival of 10.5 months in a phase II trial [43].

A meta-analysis of 24 studies (with data from 58 patients) regarding intrathecal Trastuzumab found that Trastuzumab had a tolerable safety profile, with no adverse reactions in 87% of cases and showed a significant improvement in terms of clinical response in 55% of cases. The results found a median CNS-PFS of 5.2 months and median OS of 13.2 months, concluding that intrathecal Trastuzumab can be a promising treatment option for HER2-positive breast cancer patients with LM [39].

Some trials are investigating the intrathecal use of Pertuzumab and Trastuzumab in HER2-positive LM patients [44].

In many patients, good control of CNS disease was obtained with a RR of 71.8%. The progression-free survival in patients who received IT was 15 months, and the OS was 22 months, emphasizing a meaningful and statistically significant benefit in patients treated with multimodality approaches such as systemic treatment, anti-HER2 therapy, radiotherapy, and intrathecal Trastuzumab. The safety profile was similar to previously reported studies.

Further research is needed to fully validate the efficacy of intrathecal Trastuzumab with longer follow-up and the addition of new treatment options.

Radiotherapy is a good option for patients presenting with bulky, adherent LM that are hard to target with systemic or IT therapies. It can be a great help for those with CSF flow obstruction. Typically, it is delivered as whole-brain radiation (WBRT) for cranial lesions or involved-field radiation therapy (RT) for treating the spinal cord and cauda-equina. Cranio-spinal irradiation is not recommended due to bone marrow toxicity, which can exclude the possibility of future chemotherapy regimens [7].

Focal radiotherapy can help alleviate pain by reducing the bulky masses that cause radiculopathies and can be useful in the case of obstructive lesions that cause hydrocephalus and increased intracranial pressure. By clearing obstructive lesions, RT helps increase chemotherapeutic and targeted agents’ penetration through the BBB and is, therefore, useful before systemic therapy [37,45].

However, without combining a systemic agent that targets the malignant cells disseminated throughout the CSF compartment, the therapeutic response of focal RT is limited and is associated with an increased risk of reseeding. In our lot of patients, the proportion of radiotherapy administered was 78.6% and was delivered before intrathecal administration.

Systemic disease control in HER2+ breast cancer patients has improved greatly since the introduction of anti-HER2 therapies, with Trastuzumab being the backbone of treatment for this subgroup of patients. However, with better survival rates comes a higher incidence of CNS metastases, including LM (6.8% risk of developing metastasis in 10 years). The diagnosis of LM has a devastating impact on the course of the disease, with a great reduction in the quality of life and neurologic dysfunction that impairs the ability of the patient to function independently and significantly lowers the survival rates [35,36].

Systemic treatment in our cohort reflects the actual standard of care using Pertuzumab and Trastuzumab plus docetaxel as first-line treatment, Trastuzumab emtansine (T-DM1) as the second-line treatment, and lapatinib and capecitabine or tucatinib, Trastuzumab and capecitabine as third-line treatment [46].

The novelty comes from anti-drug conjugates (ADC) like Ado-Trastuzumab Emtansine (T-DM1) and Trastuzumab-deruxtecan (T-DXd) and small molecules of Tyrosine Kinase Inhibitors (TKI) such as Neratinib and Tucatinib.

Trastuzumab emtansine (T-DM1) is a drug conjugate, incorporating the HER2–targeted Trastuzumab with the microtubule-inhibitory agent DM1. Consequent to binding onto the HER2 protein, the T-DM1 complex is internalized via receptor-mediated endocytosis. Following the internalization of the vesicle, and enzymatic processing of the complex, the T-DM1 will affect the tumor cells by combining the antitumor effects of Trastuzumab with the intracellular DM1 metabolites’ activity of disrupting the microtubule networks, leading to apoptosis and mitotic catastrophe [47,48]. The added effect of the DM1 molecule conjugation makes T-DM1 a valuable option for tumors resistant to Trastuzumab therapy. T-DM1 has proven in a phase IIIB Kamilla trial that it has good activity and is well tolerated in patients presenting with brain metastases from HER-2-positive breast cancer (overall response rate and clinical benefit rate were 21.4%) [49]. Although there was no mention of LM patients in the Kamilla trial, a few case reports and case series have shown a possible activity in patients with LM; we can therefore assume that the good CSF concentration of the drug is also active in LM lesions [50].

Trastuzumab-deruxtecan T-DXd is a newly-designed anti-HER2 humanized monoclonal antibody drug conjugated to topoisomerase-I inhibitor (Deruxtecan) with high inhibitory potency and high membrane permeability. It has shown incredible results in the DEBBRAH, TUXEDO, and DESTINY-B12 studies, proving its efficacy in multiple pre-treated metastatic HER2-positive diseases. Although T-DXd has shown a great intracranial response rate in subjects with brain metastases in both the Tuxedo-1 trial and phase II Destiny-Breast 01 study, with a median-PFS of 15 to 18 months in these patients, little is known about the efficacy in LM. This is because, like most clinical trials, LM was an exclusion criterion in both Tuxedo and Destiny B12 trials. Therefore, more clinical trials are underway to determine whether it is a good choice for patients with leptomeningeal disease—the DEBBRAH trial includes a cohort dedicated to patients with LM [51,52,53,54].

Capecitabine is an orally administered fluoropyrimidine carbamate that inhibits de novo DNA synthesis and has proven effective in treating metastatic breast cancer. A retrospective study that included three LM from breast cancer patients showed that Capecitabine has improved survival and provided clinical relief in these patients [37,55]. Lapatinib is a small-molecule TKI (with a molecular weight of approximately 500 kDa), capable of circumventing Trastuzumab-resistance mechanisms by inhibiting the intracellular kinase domain of HER-2 (contrary to Trastuzumab that targets the extracellular domain), inhibiting both EGFR and HER-2 and by accumulating HER-2 at the cell surface, leading to an enhanced Trastuzumab-dependent antibody-dependent cell-mediated cytotoxicity [14].

The combination of Capecitabine and Lapatinib has been used for BC patients with intracranial disease because they cross the blood–tumor barrier but have weak results in terms of clinical efficiency, with the Emilia trial showing that TDM-1 is superior in terms of controlling intracranial diseases in these patients [56]. A phase I trial that included a subgroup of five BC patients with LM showed that higher doses of Lapatinib in combination with Capecitabine are tolerable and may improve the oncologic outcome and intracranial response rate [57].

Other Capecitabine and TKI combinations have also been studied for patients with HER2-positive LM, such as Neratinib + Capecitabine. A small 10-patient study showed a median OS of 10 months and a CNS PFS of 4 months [58].

One study for HER2-positive metastatic disease found the combination of Capecitabine, Tucatinib, and Trastuzumab to be highly effective in treating intracranial disease. Tucatinib overcomes Trastuzumab-resistance mechanisms by selectively inhibiting the intracellular domain of HER-2, enhancing the inhibition of HER2 signaling activity alone or in combination with Trastuzumab and/or additional chemotherapy [59,60]. The HER2CLIMB trial demonstrated that adding a highly selective HER2 TKI like Tucatinib can improve the CNS-PFS of patients with brain metastases from 4.2 months to 9.9 months and increase the OS from 12 to 18.1 months. Although patients with LM were excluded from this trial, a later analysis of the drug combination was conducted in a phase 2 non-randomized study for patients with newly diagnosed, untreated LM. The median OS time in this study was nearly one year. All these new trials demonstrate that oral treatment with small molecules of TKI and anti-drug conjugates showed penetrance through the Blood–Brain Barrier (BBB) and may be a solution for these subjects [45,61,62], Table 3.

Our cohort obtained the best PFS using Tucatinib + Capecitabine + Trastuzumab (18 months) compared to Pertuzumab + Trastuzumab or T-DM1 or Lapatinib + Capecitabine, even if patients receiving Tucatinib were heavily treated and received multiple lines of chemotherapy and anti-HER2 therapy.

Interestingly, in our lot of patients, two patients underwent surgery for brain metastasis and developed leptomeningeal metastasis near the cavity of resection. A similar risk factor for LM was reported in the literature. LM incidence was higher in patients treated with surgery followed by stereotactic radiosurgery than in patients treated with radiosurgery alone [69].

The possible limitations of this study are related to the small number of patients, heterogeneity of the two groups, and limited access in the real world to all new drugs available in the guidelines. However, most trials investigating systemic treatments for Her-2-positive breast cancer excluded patients with LM metastases [49,54,62]. The number of patients we have included is comparable to other studies published on intrathecal Trastuzumab administration [39,40,41,42,43], with the added benefit of data regarding the systemic treatment administered to this very rare but very delicate group of subjects. We hope that further studies will not fail to include this group of patients and will provide even more information on the good course of action regarding survival and quality of life for Her2-positive breast cancer patients with LM metastasis.

## 5. Conclusions

Intrathecal administration of Trastuzumab, systemic treatment, and radiotherapy might improve oncologic outcomes in LM HER2-positive breast cancer with manageable toxicity. New anti-HER2 treatments, such as small molecules of TKI and anti-drug conjugates, are effective even in this sub-category of patients. To obtain the best outcome, all patients must be treated in a multidisciplinary team.

## Figures and Tables

**Figure 1 cancers-15-02508-f001:**
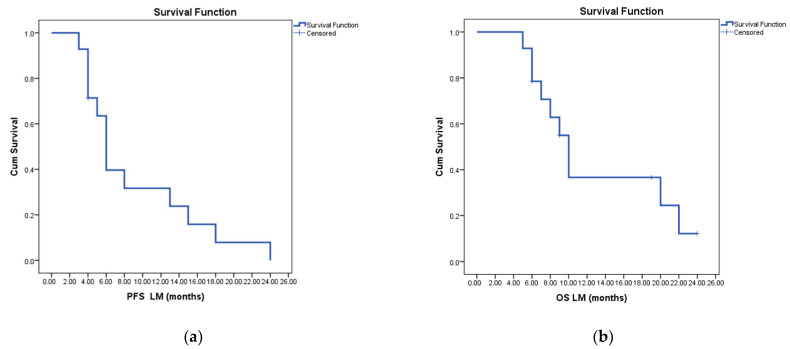
Oncologic outcomes after LM diagnosis estimated with the Kaplan–Meier method: (**a**) Median progression-free survival (6 months); (**b**) Median overall survival (10 months).

**Figure 2 cancers-15-02508-f002:**
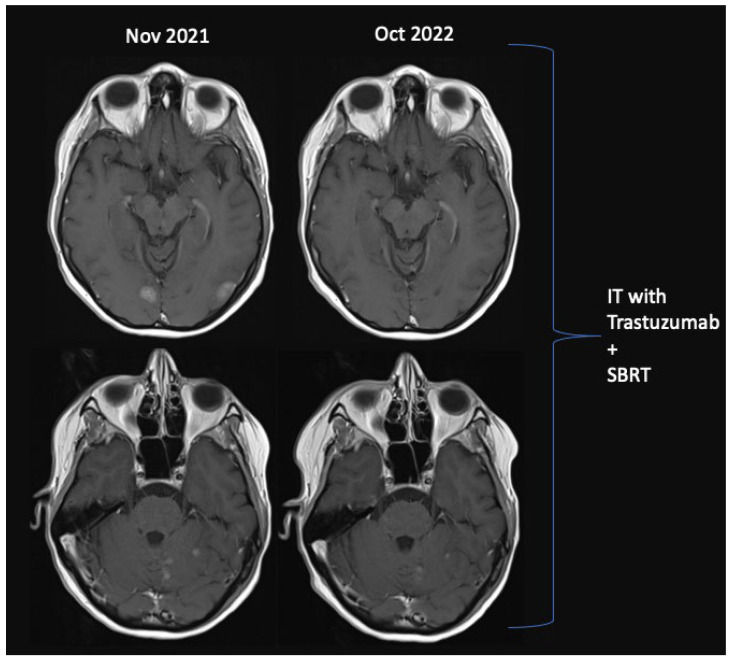
MRI showing the long-lasting response of Intrathecal Trastuzumab + SBRT + Tucatinib + Capecitabine in LM HER2-positive breast cancer.

**Figure 3 cancers-15-02508-f003:**
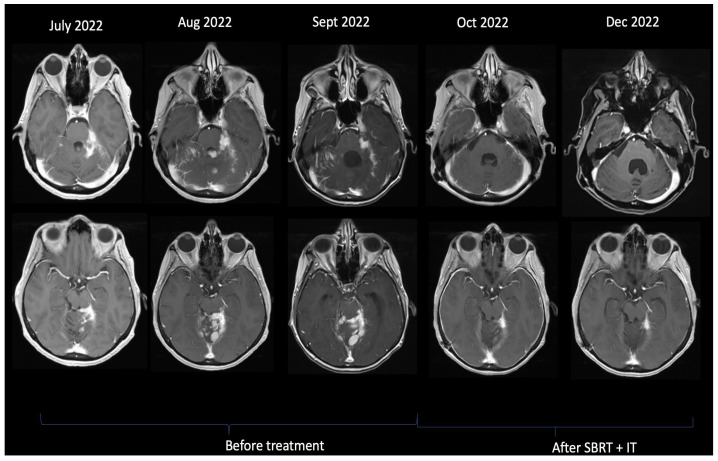
MRI showing rapid response after just two cycles of Intrathecal Trastuzumab + SBRT + Lapatinib + Capecitabine.

**Table 1 cancers-15-02508-t001:** Individual patient characteristics in the Intrathecal Trastuzumab (IT) group. KPS = Karnofsky Performance Status, BM = Brain Metastasis, WBRT = Whole Brain Radiotherapy, SBRT = Stereotactic Body Radiotherapy, LM = Leptomeningeal Metastases, PFS = Progression-Free Survival, OS = Overall Survival.

Pts	Age	KPS	Diagnosis—BM Time (mo.)	Surgery for BM	WBRT for BM	SBRT for BM	Diagnosis–LM Time (mo.)	LM Type	IT Cycles	Craniospinal RT	Systemic Therapy	PFS LM (mo)	OS LM (mo)
1.	54	100	13	Yes	Yes	Yes	41	C	12	No	Tucatinib-Capecitabine-Trastuzumab	18	19
2.	38	100	No BM	No	No	Yes	0	B	35	No	Tucatinib-Capecitabine-Trastuzumab	24	24
3.	61	60	24	Yes	Yes	No	24	C	4	Yes	Pertuzumab-Trastuzumab	4	10
4.	58	70	60	Yes	Yes	Yes	60	C	9	Yes	TDM-1	6	9
5.	67	90	110	Yes	Yes	Yes	110	A	9	Yes	Pertuzumab-Trastuzumab	15	22
6.	47	60	25	Yes	No	Yes	25	C	12	No	Lapatinib-Capecitabine	4	6
7.	48	90	48	Yes	Yes	No	48	C	4	No	TDM-1	3	6
8.	49	80	22	No	Yes	No	22	A	0	No	Lapatinib-Capecitabine	8	6
9.	43	70	122	No	Yes	No	120	C	0	Yes	Lapatinib-Capecitabine	13	20
10.	54	70	40	No	Yes	No	48	A	0	No	Lapatinib-Capecitabine	4	5
11.	72	70	12	No	Yes	No	12	B	0	No	Pertuzumab-Trastuzumab	6	8
12.	62	80	12	No	Yes	Yes	12	C	0	No	TDM-1	6	9
13.	45	70	30	No	No	Yes	40	C	0	No	Lapatinib-Capecitabine	5	6
14.	59	90	37	No	Yes	No	49	A	0	No	TDM-1	4	7

**Table 2 cancers-15-02508-t002:** Patient characteristics in the group receiving intrathecal Trastuzumab (n = 7) and in the standard of care group (n = 7).

Characteristics	Intrathecal Group	Control Group
No of patients	7 (100%)	7 (100%)
Median age at LM	54	52
KPS at diagnostic (median)	90	70
KPS 60 (%)	2 (28.6%)	0
KPS 70 (%)	1 (14.3%)	4 (57.1%)
KPS 80 (%)	0	2 (28.6%)
KPS 90 (%)	2 (28.6%)	1 (14.3%)
KPS 100 (%)	2 (28.6%)	0
Mean no of IT cycles	12.14 ± 4.00	0
Median time from BC diagnostic cu BM (mo.)	44	40
Median time from BC diagnostic to LM (mo.)	32	30
Brain metastasis (%)	6 (85.7%)	7 (100%)
Previous surgery for BM (%)	3 (42.8%)	1 (14.3%)
Previous WBRT (%)	5 (71.4%)	6 (85.7%)
Previous SBRT (%)	5 (71.4%)	2 (28.6%)
**Systemic Therapy**		
Trastuzumab + Pertuzumab + CHT (%)	2 (28.6%)	1 (14.3%)
TDM-1 (%)	2 (28.6%)	2 (28.6%)
Lapatinib + Capecitabine (%)	1 (14.3%)	4 (57.1%)
Tucatinib + Trastuzumab + Capecitabine (%)	2 (28.6%)	0
CSF+ (%)	4 (57.1%)	1 (14.3%)
MRI type A (linear) (%)	1 (14.3%)	3 (42.8%)
MRI type B (nodular) (%)	1 (14.3%)	1 (14.3%)
MRI Type C (both) (%)	5 (71.4%)	3 (42.8%)
PFS since initial BC diagnostic (months)	24	20
OS since initial BC diagnostic (months)	106	30
Median PFS since LM (months)	15	6
Median OS since LM (months)	22	8

**Table 3 cancers-15-02508-t003:** Published or ongoing trials that did not exclude patients with CNS metastasis HER2-positive breast cancers patients (RR—response rate, mo—months, AEs -adverse events, MTD—mean total dose).

Study	No Patients	Treatment	Primary End-Point
EMILIA [56]	95	Trastuzumab emtansine (T-DM1) versus Lapatinib plus Capecitabine	OS (26.8 vs. 12.9 mo)
KAMILLA (IIIb) [49]	126	Trastuzumab emtansine (T-DM1)	RR (21.4%)
DESTINY-Breast01 [63]	184	Trastuzumab deruxtecan (T-DXd)	RR (60.9%)
NALA [64]	101	Neratinib Plus Capecitabine	PFS
TUXEDO-1 [51]	15	Trastuzumab deruxtecan (T-DXd)	The intracranial RR of 73.3%
HER2CLIMB [65]	612	Tucatinib versus placebo in combination with capecitabine and Trastuzumab	PFS 9.9 vs. 4.2 mo OS 18 mo vs. 12 mo
NCT02650752 [57]	11	Intermittent High-Dose Lapatinib in Tandem with Capecitabine	Efficacy and toxicity
NCT01325207	15	Intrathecal Trastuzumab for Leptomeningeal Metastases	Dose Limiting Toxicities AEs
NCT03696030 Phase 1	39	Intraventricular administration of autologous HER2CAR T Cells	Dose Limiting Toxicities AEs
NCT04588545 [66] Phase 1/2	39	Focal RT or WBRT + Intrathecal Trastuzumab/Pertuzumab	MTD OS
NCT03501979 Phase 2 [67]	30	Tucatinib + Trastuzumab + Capecitabine	OS
NCT04420598 Phase 2 (DEBBRAH) [53]	41	Trastuzumab deruxtecan in LM	OS
NCT01494662 [68]	140	Phase 2 study of HKI-272 (neratinib) in Brain metastases	RR

## Data Availability

Not applicable.

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
