# Peer review of "Therapies beyond Physiological Barriers and Drug Resistance: A Pilot Study and Review of the Literature Investigating If Intrathecal Trastuzumab and New Treatment Options Can Improve Oncologic Outcomes in Leptomeningeal Metastases from HER2-Positive Breast Cancer"

_cancers, 2023, doi:10.3390/cancers15092508_

Round 1
Reviewer 1 Report
This study applied intrathecal Trastuzumab to HER2-positive breast cancer patients with leptomeningeal metastases (LM), which was supposed to improve oncologic outcomes by overcoming the blood-brain barrier. 14 patients with HER2-positive and LM were recruited in the study, among which 7 patients received intrathecal Trastuzumab and other 7 received the standard of care (SOC) treatment. As a result, the authors claimed that Intrathecal administration of Trastuzumab alongside systemic treatment and radiotherapy can improve oncologic outcomes in HER2-positive breast cancer with LM. Although it is an interesting clinic study, the sample number of 7 in each group is too small to obtain a reliable conclusion. Especially they received different style of systemic therapy (Table1). The age and disease course information of the recruited patients were not shown. Moreover, I am curious that the PFS and OS of patients receiving different therapies (Figure 2 and 3) showed worse outcomes than those after LM diagnosis (Figure 1).
Author Response
We are deeply thankful for your insightful feedback; your points helped us develop a better version of our manuscript! As per your recommendation to make English language modifications, we have made several changes that have been tracked. Here is a point-by-point description of the changes we have made.
Point 1: Although it is an interesting clinic study, the sample number of 7 in each group is too small to obtain a reliable conclusion. Especially they received different style of systemic therapy (Table1).
Response for Point 1: We thank the Reviewer for this intervention. Firstly, we changed the title of our article to emphasize the limitations of our study – the title changed to: “Therapies beyond physiological barriers and drug resistance: A pilot study investigating if intrathecal Trastuzumab and new treatment options can improve oncologic outcomes in leptomeningeal metastases from HER2-positive breast cancer”.
Secondly, we changed the last paragraph of our Discussions section to give a powerful argument as to why this relatively small number is, in fact, relevant to the literature:
The possible limitations of this study are related to the small number of patients, heterogeneity of the two groups, and limited access in the real world to all new drugs available in the guidelines. However, most trials investigating systemic treatments for Her-2 positive breast cancer excluded patients with LM metastases [53,58,67]. The number of patients we have included is comparable to other studies published on intrathecal Trastuzumab administration [42–44,46,75], with the added benefit of data regarding the systemic treatment that has been administered to this very rare but very delicate group of subjects. We hope that further studies will not fail to include this group of patients and will provide even more information on the good course of action regarding survival and quality of life for Her2-positive breast cancer patients with LM metastasis.
Thirdly, regarding the different types of systemic therapies, we want to state that patients followed the systemic treatment that was recommended by the ESMO Guideline at the time of the disease progression. Given the relatively small number of patients included in the study and the considerable period of time, that meant that the choice for systemic anti-HER2 treatment varied from patient to patient. We admit that a balanced study would’ve meant a comparison between same-setting, same-timing, and same-standard therapy (same drug administered at the same line of treatment, homogenized in the study group). Unfortunately, this could only work in a large, multicentric study, with patients enrolled simultaneously, following the same lines of treatment. We agree that this is a limitation of our study. However, we do consider that the data we provide is still significant for the low percentage of unfortunate patients suffering from HER-2 positive breast cancer with LM. We modified in our Results section the following paragraph:
Systemic therapy (chemotherapy and targeted anti-HER2 treatment) was administered according to the ESMO Guidelines for treating HER2-positive disease at the time of disease progression. For the entire group of patients, the systemic treatment consisted of Pertuzumab + Trastuzumab + Chemotherapy in 3 (21.4%) patients, TDM-1 in 4 patients (28.6%), Lapatinib + Capecitabine in 5 (35.7%) cases and Tucatinib + Trastuzumab + Capecitabine in 2 (14.3%) cases, both in the intrathecal group.
Point 2: The age and disease course information of the recruited patients were not shown. Moreover, I am curious that the PFS and OS of patients receiving different therapies (Figure 2 and 3) showed worse outcomes than those after LM diagnosis (Figure 1).
Response for Point 2:
Thank you for pointing that out. The median age was not different between the intrathecal and SOC. (54 vs. 52 years old), the mean age is 54.07±2.58.
Except for Tucatinib, there were no differences regarding OS and PFS according to systemic treatment. Only 2 patients received tucatinib, So the median PFS and OS (figure 1) were not very much influenced according to SPSS KM curves.
Reviewer 2 Report
In present manuscript authors have studied potential benefit of intratracheal administration of Trastuzumab in the patients of leptomeningeal metastases from HER2-positive breast cancer. Authors have concluded that intratracheal administration can improve the outcome in combination with other systemic therapies.
Concerns:
1-Authors need to discuss the potential reason for improved outcomes.
2- Figure legend should be described in more detail.
3-Figure axis labelling should be improved for better visibility.
4-Discussion section should include some alternate route drug administration studies and how they benefited the patient outcome.
Author Response
We are deeply thankful for your insightful feedback and kind words of encouragement! Your points helped us develop a better version of our manuscript. As per your recommendation to make English language modifications, we have made several changes that have been tracked. Here is a point-by-point description of the changes we have made.
Point 1: Authors need to discuss the potential reason for improved outcomes.
Response to Point 1: We would like to answer the Reviewer that the potential reason for improved outcomes has been discussed:
Lines 481-484: Intrathecal treatment is achieved by administering therapeutic agents directly into the subarachnoid space. This increases the drug concentration in the CSF, therefore inducing tumor cell necrosis. Subsequently, it is the most common way to deliver chemotherapeutic drugs in linear and non-bulky LM [38].
Lines 550 – 558: One of the main challenges is the lack of efficacy of the anti-Her2 agents in the intracranial disease because IV Trastuzumab has little to no penetration through the BBB (25-50% of cases will develop brain metastases and 6-7% will develop LM). And although the BBB can be affected by the local disease, and CNS irradiation is known to increase the barrier’s permeability, Trastuzumab still cannot reach therapeutic concentrations in the CSF [40]. In pharmacokinetics studies, Trastuzumab concentration in the CSF was 300 times lower than its concentration in the serum after IV administration [16]. Because it delivers an active medication to the location of the disease, intrathecal delivery of Trastuzumab is a desirable option for LM patients.
Point 2: Figure legend should be described in more detail. Figure axis labelling should be improved for better visibility.
Response to Points 2: Done! Thank you for the feedback!
Point 3: Discussion section should include some alternate route drug administration studies and how they benefited the patient outcome.
Response to point 3:
In our Discussions section, we included studies regarding alternate drug administration. We did re-organize the Discussions section to further emphasize the current state of the art:
Intrathecal treatment is achieved by administering therapeutic agents directly into the subarachnoid space. This increases the drug concentration in the CSF, therefore inducing tumor cell necrosis. Subsequently, it is the most common way to deliver chemotherapeutic drugs in linear and non-bulky LM [38].
Intrathecal administration of chemotherapeutic agents has proved a good approach for patients presenting with LM. However, using harsh chemotherapeutic medications such as Methotrexate, Cytarabine, and Thiotepa, administered either alone or in combination with hydrocortisone, is not without side effects, with many patients presenting with aseptic meningitis[11]. Although there are good results in preventing LM caused by leukemia and lymphoma, the role of these agents in solid tumors LM is limited because they have a narrow antitumor activity spectrum [38]. Only one study found the addition of intrathecal liposomal cytarabine to systemic treatment as an improvement in leptomeningeal metastases–progression-free survival (PFS) (3.8 vs.2.2 months) [39].
One of the main challenges is the lack of efficacy of the anti-Her2 agents in the intracranial disease because IV Trastuzumab has little to no penetration through the BBB (25-50% of cases will develop brain metastases and 6-7% will develop LM). And although the BBB can be affected by the local disease, and CNS irradiation is known to increase the barrier’s permeability, Trastuzumab still cannot reach therapeutic concentrations in the CSF [40]. In pharmacokinetics studies, Trastuzumab concentration in the CSF was 300 times lower than its concentration in the serum after IV administration [16]. Because it delivers an active medication to the location of the disease, intrathecal delivery of Trastuzumab is a desirable option for LM patients.
Multiple phase I/II trials and case reports have studied the intrathecal administration of Trastuzumab, providing a good insight into the safety and efficacy that it may provide. There has yet to be a universal agreement on the proper dosage and frequency of intrathecal Trastuzumab. However, the most common schedules used in the scientific literature range from 5 to 150 mg twice weekly, weekly, or every three weeks. Intrathecal Trastuzumab appears safe and has no known major side effects, whether used alone or in combination therapy [41–43].
Figura et al. reported 13 cases of LM treated with IT in 5 years, a mPFS of 5.6 months, and sustained response of more than six months in 2 patients [44].
One multicenter phase I/II trial has analyzed the safety and efficacy of intrathecal administration of Trastuzumab. It has been determined that intrathecal Trastuzumab can be safely administered with a dose of up to 80 mg twice weekly. The results showed stable disease in 50% of patients and partial response in 19% of cases, with a median overall survival of 10.5 months in a phase II trial [45].
A meta-analysis of 24 studies (with data from 58 patients) regarding intrathecal Trastuzumab found that Trastuzumab had a tolerable safety profile, with no adverse reactions in 87% of cases and showed a significant improvement in terms of clinical response in 55% of cases. The results found a median CNS-PFS of 5.2 months and median OS of 13.2 months, concluding that intrathecal Trastuzumab can be a promising treatment option for HER2-positive breast cancer patients with LM [46].
Some trials are currently investigating the intrathecal use of a combination of Pertuzumab and Trastuzumab in HER2-positive LM patients [47].
In our lot of patients, good control of CNS disease was obtained with a RR of 71.8%. Progression-free survival in patients who received IT was 15 months, and OS was 22 months, emphasizing a clearly meaningful and statistically significant benefit in patients treated with multimodality approaches such as systemic treatment, anti-HER2 therapy, radiotherapy, and intrathecal Trastuzumab. The safety profile was similar to previously reported studies.
Further research is needed to fully validate the efficacy of intrathecal Trastuzumab with longer follow-up and the addition of new treatment options.
Again, we thank you for your comments and suggestions,
The authors
Reviewer 3 Report
This manuscript entitled as “Therapies beyond physiological barriers and drug resistance: Can intrathecal Trastuzumab and new treatment options im-3 prove oncologic outcomes in leptomeningeal metastases from HER2-positive breast cancer?” described the results of oncologic outcomes of 14 patients with HER2-positive LM. Of them, seven patients received intrathecal trastuzumab and standard of care and 7 received the current standard of care treatment respectively.
Though the title is enchanting and attractive, but due to small size of patient numbers and heterogenous standard of care systemic treatment, the results of this study is of limited novelty and clinical significance. IN addition, the manuscript is not in good organization and the content is redundant.
Author Response
Your points helped us develop a better version of our manuscript. As per your recommendation to make English language modifications, we have made several changes that have been tracked. Here is a point-by-point description of the changes we have made.
Point 1: Though the title is enchanting and attractive, due to small size of patient numbers and heterogenous standard of care systemic treatment, the results of this study is of limited novelty and clinical significance. In addition, the manuscript is not in good organization and the content is redundant.
Response to Point 1:
We want to point out that we changed the title to “Therapies beyond physiological barriers and drug resistance: A pilot study investigating if intrathecal Trastuzumab and new treatment options can improve oncologic outcomes in leptomeningeal metastases from HER2-positive breast cancer” to highlight the niche of our study. We re-organized our manuscript’s text as to underline the importance of our results better.
Furthermore, we explained in this paragraph why the number of patients included in the study is not of small size and why the data this study provides is valuable given the current literature:
The possible limitations of this study are related to the small number of patients, heterogeneity of the two groups, and limited access in the real world to all new drugs available in the guidelines. However, most trials investigating systemic treatments for Her-2 positive breast cancer excluded patients with LM metastases [53,58,67]. The number of patients we have included is comparable to other studies published on intrathecal Trastuzumab administration [42–44,46,75], with the added benefit of data regarding the systemic treatment that has been administered to this very rare but very delicate group of subjects. We hope that further studies will not fail to include this group of patients and will provide even more information on the good course of action regarding survival and quality of life for Her2-positive breast cancer patients with LM metastasis.
Again, we thank you for your comments and suggestions,
The authors
Reviewer 4 Report
Trifanescu et al conducted this prospective design trial that assessed the added benefit by intratecal trastuzumab in her2+ metastatic breast cancer patients with leptomeningeal metastases under systemic treatment with standard of care. The idea and concept of study are very interesesting and aimed to cover an important knowledge gap in the landscape of her2+ metastatic breast cancer treatment. Unfortunately, their work presents some very relevant issues.
The most important aspect regards the number of patients included in the study, which is very low. This is unfortunately normal because the leptomeningeal metastases are quite rare and the study is monocentric. However, this does have an impact on the scientific relevance of the findings, which are invalidated by the small sample size.
Moreover, the authors did not explain how they chose whether a patient received intratecal therapy or not. Was it randomized? Were there other criteria? This must be clarified.
Another important issue regards the characteristics of the disease in the 14 patients. It is important to know ER, PR, Ki67 levels, HER2 score, whether FISH was carried out. Furthermore, it is relevant to know what was the brain metastatic burden for each patient.
An additional issue is the balance between the two groups in some important factors that determine PFS and prognosis. The authors included patients treated with pertuzumab-trastuzumab, t-dm1, lapatinib combination, tucatinib combination. Several issue arise from this fact. First, the different schedules are not balanced in the two groups. patients that received intratecal therapy, also received more tucatinib, which is more effective than lapatinib. Also, each of the mentioned regimens is given in different lines of therapy, which means patients were included in the study with different basal risk, according to the line of therapy. By the way, this aspect was not mentioned among the basal characteristics. Another unbalanced factor was performance status and surgery. Authors did not provide enough information regarding radiotherapy.
Overall, this facts limitate the statistical analysis that was conducted and its clinical significance. It is not possibile to assess the conclusions of the authors in a scientifically correct manner.
The data can be presented more appropriately as a report of a series of cases only in a descriptive fashion. No comparison between the two groups is possible.
Minor issues to be mentioned are the following:
-introduction is too long. It must be shortened and focus on the study topic. Also, the end of the introduction should "open" the study procedures.
-discussion is too long, with too many irrelevant data on studies regarding systemic therapies. It should be changed and focus on discussing the findings of the current study, compared to previous literature data.
Author Response
We want to thank you for your inspiring and insightful remarks that determined us to improve our manuscript. As per your recommendation to make English language modifications, we have made several changes that have been tracked. Here is our point-by-point description of the changes we have made.
Point 1: The most important aspect regards the number of patients included in the study, which is very low. This is unfortunately normal because the leptomeningeal metastases are quite rare and the study is monocentric. However, this does have an impact on the scientific relevance of the findings, which are invalidated by the small sample size.
Response to Point 1: We thank the Reviewer for this intervention; however historically, patients with LM Her2 positive disease have been excluded from studies regarding therapies. As such, data regarding treatment for this subset of patients is very scarce and comes from studies with a very limited number of patients. We did, however, change the title of our article to emphasize the limitations of our study – the title changed to: “Therapies beyond physiological barriers and drug resistance: A pilot study investigating if intrathecal Trastuzumab and new treatment options can improve oncologic outcomes in leptomeningeal metastases from HER2-positive breast cancer”.
Furthermore, we explain why given the current literature, the current number of patience is of reference for the literature:
The possible limitations of this study are related to the small number of patients, heterogeneity of the two groups, and limited access in the real world to all new drugs available in the guidelines. However, most trials investigating systemic treatments for Her-2 positive breast cancer excluded patients with LM metastases [53,58,67]. The number of patients we have included is comparable to other studies published on intrathecal Trastuzumab administration [42–44,46,75], with the added benefit of data regarding the systemic treatment that has been administered to this very rare but very delicate group of subjects. We hope that further studies will not fail to include this group of patients and will provide even more information on the good course of action regarding survival and quality of life for Her2-positive breast cancer patients with LM metastasis.
Point 2: Moreover, the authors did not explain how they chose whether a patient received intrathecal therapy or not. Was it randomized? Were there other criteria? This must be clarified.
Response to Point 2: In our manuscript, we documented the following information about the inclusion criteria: Inclusion criteria were HER2-positive breast cancer patients with imaging highly suggestive for LM on MRI, with or without CSF positive cytology, with Karnofsky performance status of more than 60, willing to undergo multimodality treatment and normal hematological, renal, and hepatic function.
As such, patients were considered for intrathecal administration if the treatment was feasible considering their MRI imaging having sub-type A or C lesions, their performance score and clinical fitness, and their consent for the procedure.
Point 3: Another important issue regards the characteristics of the disease in the 14 patients. It is important to know ER, PR, Ki67 levels, HER2 score, whether FISH was carried out. Furthermore, it is relevant to know what was the brain metastatic burden for each patient.
Response to Point 3:
Thank you for your comment. We up-dated the manuscript with this data in order to better characterized -the patient population. (170-173).
The histopathologic report showed invasive ductal carcinoma in all patients, estrogen or progesterone receptor positive in 10 patients (71.42%), HER2 positive (9 patients HER2 3+ on immunohistochemistry and five patients HER2 2+, positive on FISH). The mean value of Ki67 was 34.69±10.57%.
Point 4: An additional issue is the balance between the two groups in some important factors that determine PFS and prognosis. The authors included patients treated with pertuzumab-trastuzumab, t-dm1, lapatinib combination, tucatinib combination. Several issue arise from this fact. First, the different schedules are not balanced in the two groups. patients that received intrathecal therapy, also received more tucatinib, which is more effective than lapatinib. Also, each of the mentioned regimens is given in different lines of therapy, which means patients were included in the study with different basal risk, according to the line of therapy. By the way, this aspect was not mentioned among the basal characteristics.
Response to Point 4:
We thank the Reviewer for notifying this potential imbalance in our study. Patients followed the systemic treatment recommended by the ESMO Guideline at the time of the disease progression. Given the relatively small number of patients included in the study and the considerable period of time, that meant that the choice for systemic anti-HER2 treatment varied from patient to patient. We admit that a balanced study would’ve meant a comparison between same-setting, same-timing, and same-standard therapy (same drug administered at the same line of treatment, homogenized in the study group). Unfortunately, this could only work in a large, multicentric study, with patients enrolled simultaneously, following the same lines of treatment. We agree that this is a limitation of our study. However, we do consider that the data we provide is still significant for the low percentage of unfortunate patients suffering from HER-2 positive breast cancer with LM.
Systemic therapy (chemotherapy and targeted anti-HER2 treatment) was administered according to the ESMO Guidelines for treating HER2-positive disease at the time of disease progression. For the entire group of patients, the systemic treatment consisted of Pertuzumab + Trastuzumab + Chemotherapy in 3 (21.4%) patients, TDM-1 in 4 patients (28.6%), Lapatinib + Capecitabine in 5 (35.7%) cases and Tucatinib + Trastuzumab + Capecitabine in 2 (14.3%) cases, both in the intrathecal group.
Point 5: Another unbalanced factor was performance status and surgery. Authors did not provide enough information regarding radiotherapy.
Response to Point 5:
We thank you for this remark. We discussed about surgery in the following paragraphs:
Results: Four patients underwent brain surgery to eliminate brain metastasis, and two of those patients’ leptomeningeal metastasis appeared near the surgical cavity.
Indeed, not much is to discuss the role of surgery in patients with LM metastasis (if the Reviewer wishes to discuss the role of neurosurgery), given the small number of patients that have undergone neurosurgical treatment for LM.
Discussions: Interestingly, in our lot of patients, two patients underwent surgery for brain metastasis and developed leptomeningeal metastasis near the cavity of resection. A similar risk factor for LM was reported in the literature. LM incidence was higher in patients treated with surgery followed by stereotactic radiosurgery than in patients treated with radiosurgery alone [74].
Furthermore, the impact of neurosurgery hasn’t been beneficial for our patients, functioning as a risk factor rather than palliation even, which is more evidence that surgery is not a feasible option for the local therapy of LM.
Regarding radiotherapy, we stated:
In our lot of patients, there was no difference regarding PFS and OS in patients receiving WBRT and craniospinal radiotherapy. There was a statistical difference in patients receiving SBRT vs. no SBRT (mPFS 4 vs. 15 months), but data may be confounded by the fact that patients receiving SBRT had a lower tumor burden.
Point 6: introduction is too long. It must be shortened and focus on the study topic. Also, the end of the introduction should "open" the study procedures. discussion is too long, with too many irrelevant data on studies regarding systemic therapies. It should be changed and focus on discussing the findings of the current study, compared to previous literature data.
Response to Point 6: We decided to remove sections from the Introduction that we considered redundant and re-organize the Discussion section (changes are tracked). We want to point out that there are more studies regarding the systemic treatment of Her2 disease rather than local therapies of Her2-positive breast cancer LM. As such, this observation can also work as an argument as to why this research paper is so important – not much has been investigated on intrathecal Trastuzumab administration for patients with LM. We also changed the last paragraph of the Introduction section to:
“As such, we set out to demonstrate that intrathecal administration of Trastuzumab improves oncological outcomes by circumventing an important physiological barrier to treat LMs while emphasizing that the choice of systemic therapy needs to consider overcoming the mechanisms of resistance that naturally arise during the course of antiHER2 targeted therapy.” thus opening the procedures of the study more fluidly. Thank you for the suggestion!
Again, we thank you for your comments and suggestions,
The authors
Round 2
Reviewer 1 Report
The revision has a significant improvement. Several comments below have to be addressed before further consideration.
1) Quite a lot of references (such as Refs 44-47) showed clinical trials or case reports about the intrathecal trastuzumab management of HER2+ breast leptomeningeal disease. What's the novelty of the current study? Please discuss and emphasize.
2) All labels in figures need a double-check to make corrections. For example, purple line in Figure 3 represented treatment with Tucatinib + Trastuzumab + Capecitabine, but not Tucatinib only.
3) The section of Results should be reorganized to present the results in a logical manner with subheadings.
4) The section of discussion needs to be condensed.
Author Response
We thank you for allowing us to submit a revised draft of our manuscript titled “Therapies beyond physiological barriers and drug resistance: A pilot study and review of the literature investigating if intrathecal Trastuzumab and new treatment options can improve oncologic outcomes in leptomeningeal metastases from HER2-positive breast cancer.” to the Cancers Journal. We thank you and the reviewers for the insightful comments and kind suggestions, which helped us significantly improve our article. Starting from the missing details you observed, we made substantial changes and believe the issues being discussed have been substantially clarified. Please find the modifications marked in red and blue in the revised manuscript. Here is a point-by-point response to the reviewers’ comments and concerns.
Response to Reviewer 1)
Point 1) Quite a lot of references (such as Refs 44-47) showed clinical trials or case reports about the intrathecal trastuzumab management of HER2+ breast leptomeningeal disease. What's the novelty of the current study? Please discuss and emphasize.
Answer to Point 1) This study provides useful data on the intrathecal administration of Trastuzumab and systemic administration for HER2-positive breast cancer metastatic disease. This has been emphasized in the article.
Point 2) All labels in figures need a double-check to make corrections. For example, the purple line in Figure 3 represented treatment with Tucatinib + Trastuzumab + Capecitabine, but not Tucatinib only.
Answer to Point 2) We have modified our tables and figures and only included PFS and OS data for a whole lot of patients (Figure 1). We have added Table 2 to include more data about our patients.
Point 3) The section of Results should be reorganized to present the results in a logical manner with subheadings.
Answer to Point 3) We reorganized the Results section.
Point 4) The section of discussion needs to be condensed.
Answer to Point 4) We would like to preserve as much as possible the discussion section as we consider it provides a comprehensive and thorough review of the literature on a very narrow and specific topic.
Reviewer 3 Report
I appreciate the efforts from the authors. The revised manuscript is still too redundant. Intrathecal tratuzumab has been used in treating the LM clinically since 2010 with variable protocols. To match the title - "Therapies beyond physiological barriers and drug resistance. and new treatment options “, either in the section of Introduction and Discussion, the authors should focus on the rationale of intrathecal therapy when novel anti-HER2 agents like T-DxT, SG and Tucatunib are available. Besides, IT trastuzumab was administered in various doses of 25-150 mg and using different schedules, including weekly to every 3 weeks. Some experts concern about whether long-term therapy over 6 months with such a high dose is safe and well-tolerable.
Author Response
Dear Editor,
We thank you for allowing us to submit a revised draft of our manuscript titled “Therapies beyond physiological barriers and drug resistance: A pilot study and review of the literature investigating if intrathecal Trastuzumab and new treatment options can improve oncologic outcomes in leptomeningeal metastases from HER2-positive breast cancer.” to the Cancers Journal. We thank you and all the reviewers for the insightful comments and kind suggestions, which helped us significantly improve our article. Starting from the missing details you observed, we made substantial changes and believe the issues being discussed have been substantially clarified. Please find the modifications marked in red and blue in the revised manuscript. Here is a point-by-point response to the reviewers’ comments and concerns.
Response to Reviewer 3)
I appreciate the efforts from the authors. The revised manuscript is still too redundant. Intrathecal tratuzumab has been used in treating the LM clinically since 2010 with variable protocols. To match the title - "Therapies beyond physiological barriers and drug resistance. and new treatment options “, either in the section of Introduction and Discussion, the authors should focus on the rationale of intrathecal therapy when novel anti-HER2 agents like T-DxT, SG and Tucatinib are available. Besides, IT trastuzumab was administered in various doses of 25-150 mg and using different schedules, including weekly to every 3 weeks. Some experts concern about whether long-term therapy over 6 months with such a high dose is safe and well-tolerable.
Answer to Reviewer 3:
We mentioned in the introduction:
The biggest issue for the systemic treatment of HER2-positive LM patients is the need to overcome the blood-brain barrier, which is impervious to molecules with a molecular weight higher than 400-500 Da, thus limiting the efficacy of systemic treatment [21]. Unfortunately, Trastuzumab cannot pass the blood-brain barrier, having a molecular weight of approximately 148 kDa [22].
Although the blood-brain barrier can be circumvented through intrathecal administration, the concurrent systemic anti-HER2 treatment must consider that the tumor cells can become resistant to Trastuzumab during the treatment. Incriminated mechanisms of resistance include increased signaling from other HER receptors (such as HER3 or epidermal growth factor receptor) [23]; structural modifications of the antibody binding site, leading to Trastuzumab binding impairment [24,25]; mutations in the HER2/ERBB2 gene (such as L755S) [26–29]; increased intratumoral heterogeneity of HER2 expression [30,31]; and increased activity and expression of drug efflux pumps [32–34].
As such, we set out to demonstrate that intrathecal administration of Trastuzumab improves oncological outcomes by circumventing an important physiological barrier to treat LMs while emphasizing that the choice of systemic therapy needs to consider overcoming the mechanisms of resistance that naturally arise during the course of antiHER2 targeted therapy.
Details about the administration of Intrathecal Trastuzumab and safety have been added to our article (table 1). Thank you!
Reviewer 4 Report
Authors have applied significant changes to their paper. However, from my point of view the study cannot be presented with statistical analyses that compare groups since the number of patients (7 for each group) makes the statistical model not valid. Usually, the Cox model needs 10 events for each covariate. Besides the fact that the patient group is significantly heterogeneous, and systemic treatments are even more heterogeneous.
In my opionion the data obtained by this study are relevant, but they must be presented in the correct way in order to do not mislead the reader. In this case, a comparison 7 vs 7 in the modalities contemplated by the study, is not scientifically valid, in my opinonion. The most appropriate way to present the data is as a case series of 7 patients treated with intratechal trastuzumab, besides standard therapy.
Author Response
Dear Reviewer,
We thank you for allowing us to submit a revised draft of our manuscript titled “Therapies beyond physiological barriers and drug resistance: A pilot study and review of the literature investigating if intrathecal Trastuzumab and new treatment options can improve oncologic outcomes in leptomeningeal metastases from HER2-positive breast cancer.” to the Cancers Journal. We thank you and all the reviewers for the insightful comments and kind suggestions, which helped us significantly improve our article. Starting from the missing details you observed, we made substantial changes and believe the issues being discussed have been substantially clarified. Please find the modifications marked in red and blue in the revised manuscript. Here is a point-by-point response to the comments and concerns.
Response to Reviewer 4)
Authors have applied significant changes to their paper. However, from my point of view the study cannot be presented with statistical analyses that compare groups since the number of patients (7 for each group) makes the statistical model not valid. Usually, the Cox model needs 10 events for each covariate. Besides the fact that the patient group is significantly heterogeneous, and systemic treatments are even more heterogeneous.
In my opinion the data obtained by this study are relevant, but they must be presented in the correct way in order to do not mislead the reader. In this case, a comparison 7 vs 7 in the modalities contemplated by the study, is not scientifically valid, in my opinion. The most appropriate way to present the data is as a case series of 7 patients treated with intratechal trastuzumab, besides standard therapy.
Answer to Reviewer 4: We have modified our statistical analysis accordingly. We also added Table 1 to better characterize our patient’s data.
Lines 244 – 316:
Trastuzumab was administered intrathecally via repeated lumbar puncture at 150 mg every three weeks, associated or not with intrathecal dexamethasone 2 or 4mg. Descriptive statistics (mean and its standard error, median, standard deviation) were used to characterize the two groups. Kolmogorov-Smirnov normality test have been applied to test the normal distribution across the two groups of the quantitive variables (such as age, KPS, progression free and overall survivals calculated with respect to various evolution landmarks i.e. first diagnostic, brain tumor diagnostic, leptomeningeal tumor dissemination etc.). For the variables exhibiting normal distributions t-Student tests have been applied to compare the means, while for all the other we have applied non-parametric tests to compare the distributions (Mann-Whitney U-test, Kolmogorov-Smirnov test), to compare the means across groups (independent median test) and to estimate the confidence interval of median difference across the groups (Hodges-Lehmann). Since not all the variables were not normally distributed correlations coeffiecients were calculated using the Spearman scheme.
Though the groups dimensions were not very big, we attempted to classicaly evaluate the oncologic outcome for the LM patients using the Kaplan-Meier method to determine median progression-free survival (PFS) and overall survival (OS). PFS was defined as the time from LM diagnosis to the leptomeningeal disease progression on imaging or death from any cause, and overall survival was defined as the time from LM diagnosis to death of any cause. The univariate analysis using the log-rank test was used to analyze the influence of different factors regarding the oncologic outcome. A multivariate analysis was used according to the stepwise Cox proportional hazards model to identify independent prognostic factors and estimate their effect on the time to disease progression and overall survival. P value was considered statistically significant if it was < 0.05.
Table 1. Individual patient’s characteristics in the Intrathecal Trastuzumab (IT) group. KPS = Karnofsky Performance Status, BM = Brain Metastasis, WBRT = Whole Brain Radiotherapy, SBRT = Stereotactic Body Radiotherapy, LM = Leptomeningeal Metastases, PFS = Progression-Free Survival, OS = Overall Survival
Pts |
Age |
KPS |
Diagnosis – BM time (mo.) |
Surgery for BM |
WBRT for BM |
SBRT for BM |
Diagnosis – LM Time (mo.) |
LM Type |
IT Cycles |
Craniospinal RT |
Systemic Therapy |
PFS LM (mo) |
OS LM (mo) |
1. |
54 |
100 |
13 |
Yes |
Yes |
Yes |
41 |
C |
12 |
No |
Tucatinib-Capecitabine-Trastuzumab |
18 |
19 |
2. |
38 |
100 |
No BM |
No |
No |
Yes |
0 |
B |
35 |
No |
Tucatinib-Capecitabine-Trastuzumab |
24 |
24 |
3. |
61 |
60 |
24 |
Yes |
Yes |
No |
24 |
C |
4 |
Yes |
Pertuzumab-Trastuzumab |
4 |
10 |
4. |
58 |
70 |
60 |
Yes |
Yes |
Yes |
60 |
C |
9 |
Yes |
TDM-1 |
6 |
9 |
5. |
67 |
90 |
110 |
Yes |
Yes |
Yes |
110 |
A |
9 |
Yes |
Pertuzumab-Trastuzumab |
15 |
22 |
6. |
47 |
60 |
25 |
Yes |
No |
Yes |
25 |
C |
12 |
No |
Lapatinib-Capecitabine |
4 |
6 |
7. |
48 |
90 |
48 |
Yes |
Yes |
No |
48 |
C |
4 |
No |
TDM-1 |
3 |
6 |
8. |
49 |
80 |
22 |
No |
Yes |
No |
22 |
A |
0 |
No |
Lapatinib-Capecitabine |
8 |
6 |
9. |
43 |
70 |
122 |
No |
Yes |
No |
120 |
C |
0 |
Yes |
Lapatinib-Capecitabine |
13 |
20 |
10. |
54 |
70 |
40 |
No |
Yes |
No |
48 |
A |
0 |
No |
Lapatinib-Capecitabine |
4 |
5 |
11. |
72 |
70 |
12 |
No |
Yes |
No |
12 |
B |
0 |
No |
Pertuzumab-Trastuzumab |
6 |
8 |
12. |
62 |
80 |
12 |
No |
Yes |
Yes |
12 |
C |
0 |
No |
TDM-1 |
6 |
9 |
13. |
45 |
70 |
30 |
No |
No |
Yes |
40 |
C |
0 |
No |
Lapatinib-Capecitabine |
5 |
6 |
14. |
59 |
90 |
37 |
No |
Yes |
No |
49 |
A |
0 |
No |
TDM-1 |
4 |
7 |
Again, we thank you for your comments and suggestions,
The authors
Round 3
Reviewer 1 Report
My concerns have been addressed.